# Two Phages of the Genera *Felixunavirus* Subjected to 12 Hour Challenge on *Salmonella* Infantis Showed Distinct Genotypic and Phenotypic Changes

**DOI:** 10.3390/v11070586

**Published:** 2019-06-27

**Authors:** Dácil Rivera, Lauren K. Hudson, Thomas G. Denes, Christopher Hamilton-West, David Pezoa, Andrea I. Moreno-Switt

**Affiliations:** 1Escuela de Medicina Veterinaria, Facultad de Ciencias de la Vida, Universidad Andres Bello, Santiago 8320000, Chile; 2Departamento de Ciencia de los Alimentos y Tecnología Química, Facultad de Ciencias Químicas y Farmacéuticas, Universidad de Chile, Santiago 8380492, Chile; 3Department of Food Science, University of Tennessee, Knoxville, TN 37996, USA; 4Departamento de Medicina Preventiva Animal, Facultad de Ciencias Veterinarias, Universidad de Chile, Santiago 8330015, Chile; 5Escuela de Medicina Veterinaria, Facultad de Ciencias, Universidad Mayor, Santiago 8580745, Chile; 6Millennium Nucleus for Collaborative Research on Bacterial Resistance (MICROB-R), Santiago 7550000, Chile

**Keywords:** *Felixounavirus*, *Salmonella virus FelixO1*, *Salmonella virus Mushroom*, *Salmonella* Infantis, phage resistance mutants, host range, selective challenge assay

## Abstract

*Salmonella* Infantis is considered in recent years an emerging *Salmonella* serovar, as it has been associated with several outbreaks and multidrug resistance phenotypes. Phages appear as a possible alternative strategy to control *Salmonella* Infantis (*S*I). The aims of this work were to characterize two phages of the *Felixounavirus* genus, isolated using the same strain of *S*I, and to expose them to interact in challenge assays to identify genetic and phenotypic changes generated from these interactions. These two phages have a shared nucleotide identity of 97% and are differentiated by their host range: one phage has a wide host range (lysing 14 serovars), and the other has a narrow host range (lysing 6 serovars). During the 12 h challenge we compared: (1) optical density of *S*I, (2) proportion of *S*I survivors from phage-infected cultures, and (3) phage titer. Isolates obtained through the assays were evaluated by efficiency of plating (EOP) and by host-range characterization. Genomic modifications were characterized by evaluation of single nucleotide polymorphisms (SNPs). The optical density (600 nm) of phage-infected *S*I decreased, as compared to the uninfected control, by an average of 0.7 for SI infected with the wide-host-range (WHR) phage and by 0.3 for *S*I infected with the narrow-host-range (NHR) phage. WHR phage reached higher phage titer (7 × 10^11^ PFU/mL), and a lower proportion of *S*I survivor was obtained from the challenge assay. In *S*I that interacted with phages, we identified SNPs in two genes (*rfaK* and *rfaB*), which are both involved in lipopolysaccharide (LPS) polymerization. Therefore, mutations that could impact potential phage receptors on the host surface were selected by lytic phage exposure. This work demonstrates that the interaction of *Salmonella* phages (WHR and NHR) with *S*I for 12 h in vitro leads to emergence of new phenotypic and genotypic traits in both phage and host. This information is crucial for the rational design of phage-based control strategies.

## 1. Introduction

*Salmonella* is an important foodborne pathogen with global recognition [1,2]. Among the 2600 serovars, *Salmonella* Infantis (*S*I) is an emerging serovar of great significance worldwide in food production, especially in poultry production [1,2]. A recent multistate *S*I outbreak in the United States of America (USA) was linked to poultry meat [3]. In Europe, an increase in cases associated with this serovar has also been described [1]. European Food Safety Authority (EFSA) reported *S*I among the five most important serovars causing human salmonellosis in Europe [1]. In 2016, *S*I had an increase in incidence rate of 165.8% from previous years in the USA [2]. In addition, there are several reports of multidrug-resistant *S*I, including resistance to carbapenems, colistin, nalidixic acid, ciprofloxacin, tetracycline, spectinomycin, streptomycin, and sulfamethoxazole [4,5,6,7,8]. These data highlight the importance of developing innovative interventions to control *S*I. The use of bacteriophages or phages appears as a potential alternative to control this pathogen [9]. 

To date, there are important gaps in knowledge for effective phage-based application in foods [10,11,12,13]. Among these gaps is a need for information on the emergence of phage-resistant bacteria that occur through genetic mutations that are selected upon phage exposure [14]. Experimental evolutionary studies have shown that bacterial resistance to phages has been associated with modifications in their surface structures, which are used by phages as receptors [15]. One study found that all 69 characterized phage-resistant mutants of the foodborne pathogen *Listeria monocytogenes* were resistant by inhibiting phage adsorption [16]. Depending on the phage, studies have reported the development of absolute resistance or pseudo-resistance (e.g., physiological resistance) to phages [17,18]. Strains resistant to phages can have altered fitness for decreased virulence or reduced motility [19,20]. Current evidence has shown that phages are capable of rapidly responding to bacterial resistance by modifying their anti-receptors, resulting in acquired ability to recognize new host receptors [21,22,23,24]. Due to the complexity of food and animal studies, host–phage interactions have mainly been conducted in culture media [19,20]. However, there is a study that described the resistance of *Campylobacter jejuni* to phages in broiler chicken house environments [25]. The study reported resistance to phages in emerging genotypes of *C. jejuni* that are different from the ancestral population [25]. In another study conducted on *Listeria monocytogenes* and phage P100, authors investigated the effect of environmental conditions of dairy processing plants [26].

There are five phage families of the order *Caudovirales* (*Siphoviriade, Myoviridae, Podoviridae, Herelleviridae, Ackermannviridae*) [27]; among these, *Myoviridae* phages correspond to phages with contractile tails that have been reported as strictly lytic [28]. The contractile systems in *Myoviridae* have been defined as a complex system that consists of baseplate and tail fiber proteins that together are capable of introducing genetic material into the bacterial cell [28]. Within this family is the genus *Felixounavirus,* which contains several species in the genus, frequently reported as virulent representatives of *Salmonella* phages [29]. The type species is *FelixO1*, which is capable of lysing up to 98.2% of *Salmonella* strains tested [29]. These host-range characteristics have fostered research to develop phage-based interventions in food using *Felixounaviruses,* as wide host range is a desirable feature for phages to be used for biocontrol [30]. However, phages are dynamic and their host range may vary over generations [31,32,33]. Previous studies have shown that phages with a wide host range obtained from experimental evolutionary studies may be less efficient than ancestral phages [31,32,33]. Conversely, wide-host-range phages may be a better alternative as biocontrol because they could broaden their host range [15,30,34,35,36,37,38]. To develop phage-based interventions for *S*I, it is important to improve our understanding on resistance and modifications of *S.* Infantis exposed to *Felixounaviruses.* The aim of this work was to (1) characterize two *Felixounaviruses* with different host ranges, isolated from the same strain of *S*I and (2) subsequently, expose them to interact in a challenge assay, to identify resistance and genetic and phenotypic changes developed upon interactions.

## 2. Materials and Methods

### 2.1. Characterization of Host Bacteria and Bacteriophages

The strain used for this assay was *Salmonella* Infantis CHA004-4, which was isolated from a backyard poultry production system in Chile [39]. The strain was grown in tryptic soy broth (TSB, Becton-Dickinson, Franklin Lakes, NJ, USA) at 37 °C for 12–16 h. The wild-type strain (wt) referred to throughout the manuscript is CHA004-4 that has not been exposed to phage.

Two phages previously isolated on an *S*I host were selected [40]. Phage vB_Si_SF20-2 was selected to represent a wide-host-range phage (WHR) and Phage vB_Si_QUI- 1 was selected to represent a narrower-host-range phage (NHR). From now phages will be referred to as WHR phage and NHR phage. Phages were amplified on the *S*I host described above using 4 mL of Trypticase Soy Agar (TSA) 0.7%, 100 μL of phage stock and 1 mL of a 1/10 dilution of an overnight culture of *S*I. Mixtures were poured into TSA and were incubated at 37 °C for 12–16 h. Plates with confluent lysis were flooded with 10 mL of SM phage diluting and storage buffer (50 mM Tris-Cl, pH 7.5; 0.1 M NaCl; y 0.01 mM MgSO4), and SM buffer was recovered, filtered (0.25 μm), and stored at 4 °C. For this study, phages were considered wild-type (wt) in their original condition with the characteristics shown in Table 1. To evaluate the morphological characteristics of both phages, lysates were pretreated with 2% uranyl acetate [41] and transmission electron microscopy (TEM) was performed with the Tecnai /12 model, Philips equipment, using an amplification of 105,000× and a power of 80 kv. The one-step growth curve was conducted as described previously [42]. Burst size in one cycle and the latent period were calculated as previously described [42]. Experiments were conducted in three independent replicates.

*Salmonella* DNA was extracted from an overnight culture of *S*I with the DNeasy Blood and Tissue Kit (Qiagen, Valencia, CA, USA). Phage DNA was extracted as previously described using phenol-chloroform, followed by DNA precipitation with ethanol [43]. DNA concentration and quality were measured by calculating the optical density radio 260/280 with a MaestroNano Pro Spectrophotometer (Maestrogen Inc., Hsinchu, Taiwan). Library preparation and sequencing were performed by MicrobesNG at the School of Biosciences, University of Birmingham (Birmingham, United Kingdom). The genomic DNA libraries were prepared using the Nextera XT library preparation kit (Illumina, San Diego, CA, USA), and sequencing was conducted using HiSeq technology from Illumina. Reads were adapter trimmed with Trimmomatic 0.30 with a quality cutoff Q15 [44]. De novo assembly was performed using SPAdes version 3.7 [45]. *S*I and phage were further annotated with RASTtk (Rapid Annotation using Subsystem Technology version 2.0; http://rast.theseed.org/FIG/rast.cgi) [46]. A complete alignment of the nucleotide sequences of both phage genomes was carried out through BLASTn [47]. EasyFig software version mc 2.1 [48] was used to construct a linear representation of phage genomes. The NCBI BioSample accession number for the *S*I used in the assay is SAMN11352530 and GenBank accessions for the phages are listed in Table 1.

### 2.2. Selective Challenge Assay of SI Exposed to WHR and NHR Phages

Wild-type *S*I from the frozen stock was grown in TSB for 16 h, then it was adjusted to an optical density OD_600nm_ (OD) of 1.0. Individual phage infections were conducted with 1 mL of *S*I OD of 1.0 (6 × 10^6^ CFU/mL) and 1 mL of phage lysate at approximately 2 × 10^4^ PFU/mL, this assay was conducted at a multiplicity of infection (MOI) of 0.01. These mixtures were incubated at 37 °C with shaking at 100 rpm for 10 minutes. Subsequently, the mixtures were centrifuged at 12,000 g × 10 min, and the pellet was resuspended in 10 mL of TSB and incubated for 12 h at 37 °C with shaking at 100 rpm [49]. Four biological replicates were conducted for each challenge assay of *S*I exposed to WHR phage and NHR phage; additionally, every hour, three variables were measured: (i) phage lysis capacity through OD [50], (ii) proportion of *S*I survivors from phage-infected cultures, and (iii) phage titer. To measure the OD, 100 uL of the mixture was obtained, and the OD was measured using a spectrophotometer (C40, Implen–ALE, Munich, Germany). To obtain the proportion of *S*I survivors from phage-infected cultures, mixtures were diluted 10^−6^ in 0.8% of NaCl, and 100 uL of this dilution was mixed with 4 mL of TSA 0.75% and plated onto TSA for a double layer. Overlays were incubated at 37 °C for 12–16 h, as previously described [49,50].
(1)$$\mathrm{The}\;\mathrm{proportion}\;\mathrm{of}\;SI\;\mathrm{survivors}\;\mathrm{from}\;\mathrm{phage}-\mathrm{infected}\;\mathrm{cultures}\;\mathrm{was}\;\mathrm{calculated}\;\mathrm{each}\;\mathrm{hour}\;\mathrm{as}:\;\frac{\mathrm{concentration}\;\left(\mathrm{CFU}/\mathrm{mL}\right)\;\mathrm{of}\;SI\;\mathrm{survivors}}{\mathrm{concentration}\;\left(\mathrm{CFU}/\mathrm{mL}\right)\;\mathrm{of}\;SI\;\mathrm{control}\;},$$ according to previous reports that calculated the proportion of phage-resistant mutants [49,50]. In each replicate, 10 colonies of *S*I survivors were selected, and this population of pooled survivors was stored in glycerol (30%) at −80 °C. To measure the phage titer, 100 uL of each mixture was centrifuged at 12.000*g* × 10 min, to remove the cells in the mixture. Then, the phages were separated by filtration of the supernatant using a filter of 0.22 µm in chloroform (1%) at 4 °C (Figure 1).

PCR was used to ensure phage and *S*I were adequately separated. Two sets of primers were designed from the nucleotide sequences of both phages. For the WHR phage, primers were synthesized to target an *HNH homing endonuclease encoding gene*, F-5′ GTGGGCTACCGAAAGGTGTT 3′ and 5′R- TATGGGCTTCTTCAGGGGTG 3′; its amplicon was 252 bp. For the NHR phage, the *Immunoglobulin I-set encoding* gene, F-5′GGTGAAGGTGGCTCAAGTGT3′ and R- 5′CAGCGGTTGCAC3′ was used, and an amplicon of 378 was obtained. For *S*I, *invA* PCR was used, as previously described [51].

### 2.3. Statistical Analysis

To analyze if the differences in the *Salmonella* optical density upon challenge assays were statistically significant, a nonparametric statistical analysis was conducted with (Kruskal Wallis, *p* < 0.05) [52]. The differences in phage titer and proportion of *S*I survivor to phages were analyzed through (ANOVA, *p* < 0.05). Analyses were conducted using the statistical software Infostat, released 2016 [53].

### 2.4. Evaluation of Phage Sensitivity by Efficiency of Plating on Isolated SI Survivors

To determine sensitivity of *S*I survivors, every hour, efficiency of plating (EOP) assays were conducted. Isolates obtained from each of the four replicates of the challenges were successively inoculated onto TSA for two passages or until the absence of lysis plaques. EOP was performed by using the double-layer overlay method [18,54], in which TSA plates were incubated at 37 °C for 18 +/− 2 h. These assays were conducted in duplicate. Results were expressed on a scale of complete resistance (0) to complete sensitivity (1), as previously reported [54]. The EOP assay was calculated by the ratio of the average PFU/mL (wild-type WHR and NHR phage) on a target host, and the population of *S*I survivors from WHR and NHR phage challenge assays, to the average PFU/mL (wild-type WHR and NHR phage) on a corresponding reference host (wild-type *S*I), as previously described [54]. 

### 2.5. Efficiency of Plating on SI Survivors Obtained from Cross-Resistance Assays

EOPs were used to evaluate whether *S*I exposed to phages obtained from the two assays (with WHR and NHR) showed similar phage sensitivity to the other phage. EOP, as described above, was conducted on *S*I survivors to WHR phage using NHR phage and *S*I survivors to NHR phage using WHR phage. EOP was conducted in duplicates for each *S*I obtained from each of the four replicates. 

### 2.6. Genotypic Evaluation of Changes in SI and Phages Obtained from the Selective Challenge Assay

To evaluate genotypic changes in *S*I survivor and phages obtained at 12 h, we used whole genome sequencing. For WGS we selected survivors from replicate one, which represented the greatest difference in proportion of *S*I survivors, which could provide evidence of genetic changes. For *S*I, we sequenced (1) wild-type *S*I, (as described in Section 2.1), (2) *S*I from the control at 12 h, (3) the population of pooled *S*I survivors (comprised of 10 colonies) to WHR phages obtained at 12 h from replicate one, and (4) the population of *S*I-pooled survivors (comprised of 10 colonies) to NHR phage obtained at 12 h from replicate one. For phages, we sequenced (1) the lysates of wild-type WHR and NHR phages, (as described in Section 2.1), (2) the lysates of WHR and NHR phage controls at 12 h, and (3) the population of WHR and NHR phages from the selective challenge assays obtained at 12 h from replicate one, as described above. DNA purifications and sequencing for *S*I and phage lysates were conducted as described above. *S*I and phage isolates from the selective challenge assays were separated in *S*I stocks and phage lysates (as explained in Section 2.3). Purified DNA from *S*I was analyzed by PCR to verify the absence of DNA from WHR and NHR phages, using the primers designed to target phages described in Section 2.3. Conversely, phage DNA was analyzed for the absence of *S*I DNA using PCR targeting *invA*. Raw reads for the 12 h samples of *S*I and phages were uploaded to NCBI SRA under BioProjects PRJNA531205 and PRJNA533931, respectively.

The first comparison of the phage genomes was made through multiple alignments between WHR and NHR phages. In order to identify genomic changes, we used McCortex version 0.0.3 for variant calling, for which *S*I and phage paired-end reads were trimmed with Trimmomatic version 0.35 [44] and quality-checked with FastQC version 0.11.7 [55]. Reads from wild-type *S*I and phage genomes were assembled with SPAdes version 3.12.0 [45], and assembly statistics were generated using BBMap (BBTools version 38.08) [56], SAMtools version 0.1.8 [57], and Quast version 4.6.3 [58]. Contigs were annotated with RASTtk (Rapid Annotation using Subsystem Technology version 2.0; http://rast.theseed.org/FIG/rast.cgi) [46]. For wild-type phages, we re-oriented the assembled genomes to begin at the large terminase subunit.

To avoid considering a spontaneous mutation as phage-induced mutation [18], a time control *S*I (*S*I unexposed to phages that was cultured with identical condition as *S*I exposed to phages for 12 h) was also sequenced and analyzed. This control allowed us to account for mutations that occur spontaneously in *S*I. An equivalent time control was also used for the WHR and NHR phages. The McCortex [59] pipeline was used with the “vcfs” argument (using links and joint calling) for graph-building and variant-calling using both the bubble caller and breakpoint caller. For *S*I samples, kmer size of 103 was used (optimal kmer size was determined using KmerGenie [60]). The wild-type *S*I assembly was used as the reference. A kmer size of 31 was used for WHR phages, and 81 for NHR phages (optimal kmer size was determined using KmerGenie [60]). The wild-type phage assemblies were used as the references. 

### 2.7. Phenotypic Evaluation of Changes in the Host Range of WHR and NHR Phages, Obtained from the Selective Challenge Assay

To determine phenotypic changes in phages obtained from selective challenge assays, host-range characterization was conducted, as previously described [40], using 25 isolates representing 23 different serovars. Presence of lysis was scored, as previously described [40]. We compared the wild-type WHR and NHR phages with phages obtained in each replicate upon the 12 h assays. This assay was also conducted in duplicate.

## 3. Results and Discussion

In this study we report the genomic and characteristics of two newly sequenced phages infecting *S*I, one with WHR and one with NHR. Phages were subjected to 12 h selective challenge assays on *S*I to compare phenotypic and genomic changes induced by the interactions. Main findings in this study include: (i) two highly similar phages present distinct lytic and phenotypic characteristics, (ii) the two phages showed significantly different effects on *S*I OD and titer after exposure, (iii) *S*I and phages showed different genotypic responses to selective challenge assays, and (iv) phenotypic modifications were greater in the WHR phage than in the NHR phage interactions with *S*I.

### 3.1. Two Highly Similar Phages Present Distinct Genotypic and Phenotypic Characteristics

Genomic, taxonomic, and phenotypic characteristics of the selected WHR and NHR phages are shown in Table 1. The morphological characteristics of the phages obtained through TEM (Appendix A) showed that both phages correspond to the order *Caudovirales,* of the *Myoviridae* family, which is characterized by having a contractile neck and with common characteristics of the genus *Felixounavirus* [61]. 

A comparison using the BLASTn algorithm of these phage genomes showed that the WHR phage’s top BLAST hit is a phage previously sequenced, named *Salmonella virus Mushroom,* with GenBank acc: KP143762.1 [62], and the NHR phage’s first BLAST result is *Salmonella phage FelixO1,* with GenBank acc: AF320576.1 [61]. A BLASTn comparison between WHR and NHR phages showed 97% nucleotide identity, shared among their genomes. Figure 2 shows the name of the genetic product encoding each gene, the function, the nucleotide and amino acid size, the percentage of GC (GC%), and the percentage of nucleotide homology. Main differences between these two phages were found in seven genes that encode tail protein: gp51 T4-like, hypothetical protein, gp10 T4-like, gp36 long tail fiber T4-like, homing endonuclease 1, tail protein gp38 T4-like, and homing endonuclease 2. In S16-T-even phage [30], it constitutes the full-length long tail fiber (LTF), which apparently represents a trimer of gp37 with gp38 as adhesin attached to its distal end. It has also been identified the putative function of gp57A tail fiber trimerization chaperone, but is not required for synthesis of functional LTF. 

Long tail fiber protein type gp37-like is known to be associated with a broad specificity by hosts [30,34,35,36,37,38]. Despite their genomic similarity, these phages displayed different host ranges; WHR phage was able to lyse 14 different *Salmonella* serovars and NHR was able to lyse six different serovars (Table 1). While *Salmonella* serovars lysed by WHR phage represent four distinct serogroups, serovars lysed by NHR phage represent only two; this could indicate that WHR phage may recognize more than one receptor in *Salmonella.* For this phage genus, the liposaccharides have been reported as the main *Felixounavirus* phage receptor. [63,64,65]. A previous study reported that three *Felixounavirus* phages isolated from dairy cows showed different host ranges [66]. This, together with our data, may indicate that there is a great diversity in the host range of the *Felixounaviruses*, which could be associated with modification in the genomes, for instance in the baseplate and tail proteins, whose variation has been associated with modifications in the host range [21,22,67]. 

Characterization of the one-step growth curve showed additional evidence of phenotypic differences (Table 1). While WHR phage showed a larger burst size with 30 particles released in the first cycle, NHR phage released only 12.6 particles, indicating a higher lytic capability of WHR phage for the *S*I used in this study. Great variability in burst size in *Felixounavirus* has been reported in other studies: between 11–116 viral particles and the period of latency of approximately 20 min [68,69]. These results indicate important diversity of phages in this genus.

### 3.2. The Two Phages Showed Significantly Different Effects on SI OD and Titer Post-Infection

The phage lytic capacity was estimated through the comparison of the OD_600nm_ reduction of *S*I challenged with WHR phage, NHR phage, and the control without phages. This parameter was previously used as an indicator of bacterial death kinetics (killing curve) in *Salmonella* Typhimurium and *Felixounaviru*s [50]. Our results showed that the WHR phage exhibited a significant reduction in the OD, in comparison with the control and NHR phage (*p* < 0.05), at all sampled times (Figure 3A). However, NHR phage only showed a significant reduction in comparison with the control in the first three hours (Figure 3A). Between both phages (WHR and NHR), nonsignificant differences were found in the first three hours, but after this time, both assays showed significant differences in the OD measurements.

The average reduction for WHR was 0.6 points of OD, whereas for NHR it was 0.3 points. In the challenges, in broad terms considering the four replicates the titers obtained were also different in WHR and NHR phages, the WHR phage showed a significantly higher titer than the NHR phage (*p* < 0.05) (Figure 3B). In the first six hours no significant differences between the titer of WHR and NHR phage were consistently observed, and then the titer of NHR phage increased until reaching 10^7^ or 10^8^ PFU/mL (Figure 3B), whereas the WHR phage continued to increase after 7 to 12 h, until reaching a titer of up to 7 × 10^11^ PFU/mL. These results demonstrate that, under the same conditions (host strain, culture medium, time of exposure, MOI, temperature, and shaking), the WHR phage showed increased lytic capacity and higher titer than the NHR phage. Interestingly, the WHR phage showed a larger burst size, which could be related with its ability for growth at higher titers in this *S*I strain. This could be consistent with a previous report [70], which generated a model using delay-differential equations to predict the interactions between a broad-host-range *Salmonella* phage and its pathogenic host and demonstrated that the parameters that play a major role for the lytic capacity of a host-phage-model correspond to latency period and burst size.

The results of the proportion of *S*I survivors obtained in each time are shown in Table 2. The average proportion of *S*I survivors after exposure to WHR was 2.7 × 10^−6^, and for *S*I exposed to NHR was 7.6 × 10^−6^. An increase in proportion was observed in *S*I exposed to NHR between 2–3 h, reaching 4 × 10^−5^. This proportion increase in NHR phage corresponded to a slight decrease in the OD (lower lytic capacity) of this phage’s challenge culture from hour 2–3. A recent study reported, using a similar methodology, the isolation of bacteria potentially resistant or pseudo-resistant [18]. However, authors reported that when using a challenge of *Bacillus thuringiensis,* with a MOI of 1.000 and exposure of 12 h, up to three of nine bacteria became pseudo-resistant [18]. They indicated that nonspecific lysis of phage-resistant cells could occur from endolysins produced by infected susceptible cells. In the present study, this is unlikely because the MOI used was much lower. In addition, we subsequently tested the *S*I exposed to phages through EOP assays. Overall, the three parameters measured to compare the efficacy of both phages showed that the WHR phage may possess qualities that indicate it could be a better option for biocontrol of *S*I.

### 3.3. SI and Phages Generated Different Genotypic Responses to the Selective Challenge Assays

The importance of performing a variant analysis was to demonstrate that during a short exposure time of 12 h, genetic changes developed in both *S*I and phages. In addition, this analysis allowed us to confirm that these genetic changes in *S*I are located specifically in genes that are responsible for the synthesis of structures which could function as receptors for phages of the *Felixounavirus*. For phages, more mutations in different genes were found in WHR phages than in NHR phages, and in WHR phages, mutations are located in genes encoding Baseplate J-like and gp37-like proteins. According to the variant filtering criteria mentioned in Section 2.6, we compared all identified variants in the 12 h exposed samples to those identified in the *S*I and phage (WHR and NHR) control strains. This consideration allowed us to filter out all spontaneous mutations occurring over the time of exposure, as previously described [18,71]. In both 12 h-exposed phage genomes, we identified subpopulations with and without mutations in the Baseplate J-like gene, and in the 12 h time-control genomes, subpopulations with and without the mutations were observed for all genes listed in Table 3. Conversely, genomes of *S*I isolates exposed to WHR and NHR phages and time-control isolates did not indicate the presence of subpopulations (i.e., the two variants identified were present at 100% frequency in the exposed isolate genomes and 0% frequency in the time-control genomes).

Our results showed that *S*I exposed to the WHR phage contained a nucleotide deletion in the gene *rfaK* that encodes for UDP-glucose: glucosyl lipopolysaccharide alpha-1,2-glucosyltransferase, an enzyme that drives the polymerization of the lipopolysaccharide (LPS) core [72] (Table 3). This deletion causes a frameshift that leads to a premature stop codon downstream, which would result in a truncated protein (Figure 4). For *S*I exposed to NHR phage, an SNP was observed in the *rfaB* gene, which encodes the UDP-D-galactose-Lipopolysaccharide 1,6-galactosyltransferase [73]. In this case the effect on the protein was a radical nonsynonymous substitution (the neutral amino acid glutamine would be substituted with the basic amino acid lysine; Figure 4; Table 3). In both cases, the variants showed a 0% frequency in the control and a 100% frequency in the *S*I exposed to phages for 12 h. These results indicate that the mutations occurred in different genes of *S*I exposed to WHR and NHR phages, but that both of these mutations could affect their protein products. Interestingly, it has been demonstrated through the generation of mutants of *Salmonella enterica* serovar Typhimurium comprising *rfaK, rfaB, rfaG, rfbP, rfbN, rfbU, rfbH,* and *rfbA* that these mutants were unable to colonize intestines of the calf [74]. This, along with our findings, could indicate that LPS mutants emerged by phage interaction, which may have decreased invasion abilities. Importantly, the O-antigen and core polysaccharide portions of the LPS are known *Felixounavirus* receptors [63,64,65]. It has also been previously shown that phage adsorption is affected by the structure and composition of the LPS [64]. The two *S*I genes with mutations identified in the present study encode enzymes that are responsible for transferring glucosyl/galactosyl residues to LPS [72,73], so mutations rendering them unfunctional or affecting their functionality would likely affect LPS structure. Thus, phage adsorption could be inhibited. This is consistent with observations that mutations in genes whose enzyme products are responsible for synthesizing the LPS core have been shown to affect the structure of the LPS core chains, which confers resistance to phage *Felix O1* [63,64,65]. Our results show that *S*I likely develops phage resistance by accumulating mutations that affect phage receptors on the host’s cell surface. This is consistent with a number of studies showing that phage resistance often develops by this means [18]. 

In response to the development of phage resistance in *S*I, we observed several mutations in the phage genomes after 12 h exposure, presumably to overcome this resistance. Many of these mutations are in genes that encode phage proteins that mediate phage adsorption (i.e., tail fibers and baseplate [18,30]). In particular, it has been demonstrated that synergism mutation of baseplate-associated proteins was essential for the phages regained infectivity [18,30]. While the analysis of *S*I showed clear variant frequency in the *S*I exposed to phages, in the analysis of control versus exposed phages we identified potential subpopulations of phages. These originated either from the stocks of wild-type phages used in the challenges or during the amplifications conducted to isolate enough DNA for sequencing. Importantly, the variant analysis showed that the WHR phage had multiple variants in each of two genes (phage protein and long tail fiber gp37-like; Table 3), which was not seen for the NHR phage. This could indicate that the WHR phage showed greater capacity for genetic modifications, as it was exposed to the same conditions as the NHR phage. In the case of the gene coding for baseplate J-like protein in the WHR phage, the variant had greater frequency in the control than in the exposed phage, and the amino acid substitution was nonsynonymous but conservative (as glycine and valine are both neutral, nonpolar amino acids; Figure 5). Conversely, the variant identified in NHR’s baseplate J-like protein gene had a frequency of 53.33% in the control and a frequency of 77.27% in the phage obtained after 12 h exposure, and the mutation leads to a radical nonsynonymous substitution that could have a protein effect (as the acidic amino acid glutamate would be substituted with the basic amino acid lysine; Figure 5).

For the long tail fiber protein gp37-like, we observed variants in both the control and in WHR phage exposed for 12 h, but with a higher frequency in the exposed phage. One of these mutations would cause a radical nonsynonymous substitution (the neutral amino acid glutamine would be substituted with the basic amino acid lysine; Figure 5), and the other was a conservative nonsynonymous substitution (leucine would be substituted with proline, both of which are neutral and nonpolar; Figure 5). Interestingly, the proline that was substituted in the second mutation was the same amino acid present at the same position in that gene in the NHR phage when the amino acid sequences were aligned (Figure 5). A study reported that the T4-like *Salmonella* bacteriophage vB_SenM-S16 of the *Myoviridae* family has a binding specificity mediated by the long tail fiber proteins [30]. In addition, Chen et al. [37] verified that alterations in tail protein gp37 expanded the host range of a T4-like phage. The gp37 protein has been identified to have a great affinity for binding different receptors in *Salmonella* [30]. The mutation obtained in WHR at 12 h was located in amino acid 497 and 534 (Figure 5). A previous report [78] described the C-terminal segment of this protein as responsible for LPS binding. This may indicate that this SNP is not necessarily associated with the changes on the host range observed here. However, it has been documented that the expansion of the host range indicated that regions different from the C-terminal region might indirectly change the specificity of the receptor by altering the supporting capacity of the binding site [22,37]. Though, one single nucleotide polymorphism (SNP) between WHR and NHR phages at position 782 (Figure 5) could be associated with the host range and needs to be further investigated. To date, a higher host-binding effectiveness has been studied when phage-chaperone proteins are present; they facilitate the folding of gp37 in T4 phages [79,80]. In future studies, it would be interesting to recognize the presence of gp37 chaperone proteins that could facilitate binding in WHR phage.

The aforementioned mutations were informative as to the role they may serve in phage-binding and overcoming phage resistance. However, our results also showed mutations in two genes of which the function was unknown, so the role these genes may serve in phage infection is unclear. We observed two distinct mutations in a gene annotated as a phage protein in the WHR phage. One of these mutations would lead to a radical nonsynonymous mutation (the neutral, nonpolar amino acid alanine would be substituted with the acidic, polar amino acid glutamate) and the other to a conservative nonsynonymous mutation (alanine would be substituted with the neutral, nonpolar amino acid valine; Appendix A). These mutations were seen in varying frequencies in both the control and exposed phage genomes. Similarly, we observed a mutation in a gene annotated as a hypothetical protein in the NHR phage, which would lead to a radical nonsynonymous mutation (the polar amino acid serine would be substituted with the nonpolar alanine; (Appendix A). However, this mutation was only observed in the control genome.

### 3.4. SI and Phages Showed Different Phenotypic Responses to Selective Challenge Assays

The EOP obtained from *S*I exposed to WHR phage showed diversity in each of the four replicates, with an average 0.3, however, common lysis patterns can be seen for isolates obtained from WHR assay when exposed to WHR and NHR phages (Table 4, Appendix A). This indicates that *S*I exposed to WHR phage acquired resistance in some of the replicates. Conversely, in *S*I exposed to NHR phage, the EOP was 0 in all isolates from all four replicates tested. *S*I interacted with NHR phage showed resistance to the wild-type NHR phage (Table 4; Appendix A). The great diversity identified in each replicate for *S*I exposed to WHR phage may indicate that NHR phage and WHR phage induced different outcomes in *S*I exposed to them. In a previous study, three of nine bacterial isolates in a selection challenge with phages were pseudo-resistant, which could be explained by the random selection of colonies from the total population, as described by Yuan et al. [18].

In the cross-resistance assay, we found an average EOP of 0.5 in *S*I exposed to WHR phage when this SI was challenged with wild-type NHR phage (Table 4, Appendix A). However, we found diversity in the replicates, as described above for *S*I exposed to WHR phage. *S*I exposed to NHR phage challenged with wild type NHR phage showed a consistent EOP of 0 (Table 4, Appendix A). Our results showed that although these two phages are highly similar, they may induce different responses in the same *S*I. This indicates that *S*I exposed to NHR phages induces cross-resistance to WHR and NHR phage, possibly due to a continuous interaction of *S*I with NHR-induced modifications of a common receptor, which allows the recognition of WHR and NHR phages. Conversely, *S*I exposed to WHR does not consistently induce cross-resistance, which suggests that WHR phage could be better able to change. Some of the possible mechanisms involved include that WHR phage rapidly modifies its anti-receptor, restriction activation, modification of DNA replication, or decreased transcription of proteins used as phage receptor [15]. Our data is consistent with data reported by Wright et al. [81], who have demonstrated through a laboratory experimental evolution of resistance against 27 phages in *P. aeruginosa*, that cross-resistance is most common against multiple phages that use the same receptor. Authors strongly recommended conducting these studies to ensure a successful application in phage therapy. Considering the EOP assay results, it is possible to hypothesize that WHR phage may possess the ability to rapidly modify its anti-receptor, or it could recognize more than one receptor in *S*I [30]. Conversely, NHR phage could recognize a single receptor, then resistance could occur more rapidly in *S*I exposed to NHR phage.

Previous reports have shown that similar phages select different cell surface structures or may differ in binding intensity (i.e., reversible and irreversible) [30]. Our findings demonstrated acquisition of cross-resistance to WHR phage upon exposure to NHR phage, emphasizing the importance of characterizing cross-resistance, especially when phages are aimed to be used for biocontrol in cocktail formulations [81].

We compared the host range of the wild-type WHR and NHR phages with phages obtained in each replicate of the 12 h assays. Important variability was obtained between the replicates on WHR phage (Table 5). Whereas in replicate one an increase of the host range from 14 *Salmonella* serovars lysed to 16 *Salmonella* serovars lysed was observed; in replicates two, three, and four the host range decreased to 12, four, and four *Salmonella* serovars, respectively (Table 5). Importantly, for WHR phage, variability among replicates was also found in the EOP assays. Differently, the NHR phage assay showed minor differences on the host range of phages upon the 12 h assays, with minor changes to the *Salmonella* serovars lysed. Importantly, we also observed major changes in the genomes of WHR phage upon 12 h of interaction with *S*I. These changes in the genome of WHR could partially explain the findings.

It is important to emphasize that in the four replicates of *S*I challenge with NHR phage, it consistently lost the ability to lyse wild-type *S*I host, which may indicate that SI changed to avoid infection with wt NHR phage. This situation was not observed with the WHR phage, since WHR phage continued to lyse SI in all four replicates (Table 5). This information coincides with the resistance found in the EOP assay, in which all four replicates showed resistance.

Considering the information obtained from the genetic variant analysis and the profile of susceptible *Salmonella enterica* serovars obtained through the host-range assay, for each phage (WHR and NHR), the presence of the *rfaB* and *rfaK* genes in each susceptible serovar was investigated (Appendix A). While this was conducted to identify if the serovars lysed by WHR and NHR phage contained the genes that we identified with SNPs [82]. In the case of the WHR, 21 of the 23 serovars tested were susceptible to this phage (Table 5) in at least one of the replicates. In genomes available at NCBI, the presence of annotated *rfaB* was found in most of the serovars previously mentioned. The *rfaK* gene has been described in all serovars susceptible to WHR, except serovars Choleraesuis and Corvallis, which belong to serogroups O:7 (C_1_) and O:8 (C_3_), respectively (Appendix A). From the profile of susceptible serovars obtained from NHR phage exposure, 11 *Salmonella* serovars of the 23 tested were susceptible. The *rfaB* gene has been described in all serovars susceptible to NHR, except serovars Muenster and Choleraesuis, which belong to serogroups O:3,10 (E_1_) and O:7 (C_1_), respectively (Appendix A). Additionally, serovars susceptible to WHR phage represented more serogroups than serovars susceptible to NHR phage. In both cases the most common serogroup was O:7 (C_1_), which corresponds to the same serogroup than *Salmonella* Infantis. It is then further necessary to study if WHR phage recognizes more than one receptor, whereas if NHR can recognize only one putative receptor. Future studies could identify whether there is an association between SNPs in *rfaB* and *rfaK* and phage infection. Furthermore, site-directed mutation in *rfaB* and *rfaK* genes could provide additional evidence to better understand the roles of the SNPs found in these genes and how they affect the host range of the studied phages. A study described by Marti et al. [30] demonstrated that simple mutants of OmpC (ΔompC) and LPS outer core (lpsRe) were not effective in forming strains completely resistant to infection by S16, but the double mutants of these genes were effective. This is because gp37, located at the tips of the long tail fibers, reversibly binds to LPS or OmpC [30]. This first adsorption activates a second adsorption, in which gp12 (short-tailed fiber) irreversibly binds to LPS on the cell surface [30]. This could indicate that it is necessary to study additional prospective phage receptors in order to characterize WHR phage and its potential use for biocontrol of *S*I.

Overall, our results show that although the WHR phage showed a greater lytic capacity than the NHR phage, the WHR phage presents the largest number of mutations leading to nonsynonymous amino acid substitution and variability in host range. In the WHR phage studied here, mutations were seen in the gene encoding long tail fiber gp37-like protein. It has been described previously that modification in this protein could change the host range of the phage [30,34,35,36,37,38], which could explain our findings in the greater diversity found in host range in the WHR phage during the four replicates. Although the results obtained represent the interaction of two WHR and NHR phages against a single strain of *Salmonella* Infantis, these results show the importance of conducting phage–host challenges in order to evaluate viral titer, phage-resistant mutants, host range, and genetic changes. All of these factors are necessary to evaluate in order to design rational and effective phage-based biocontrol applications. It should be noted that this study was conducted under experimental conditions in vitro, which may not represent *Salmonella* phage interactions in nature.

## Figures and Tables

**Figure 1 viruses-11-00586-f001:**
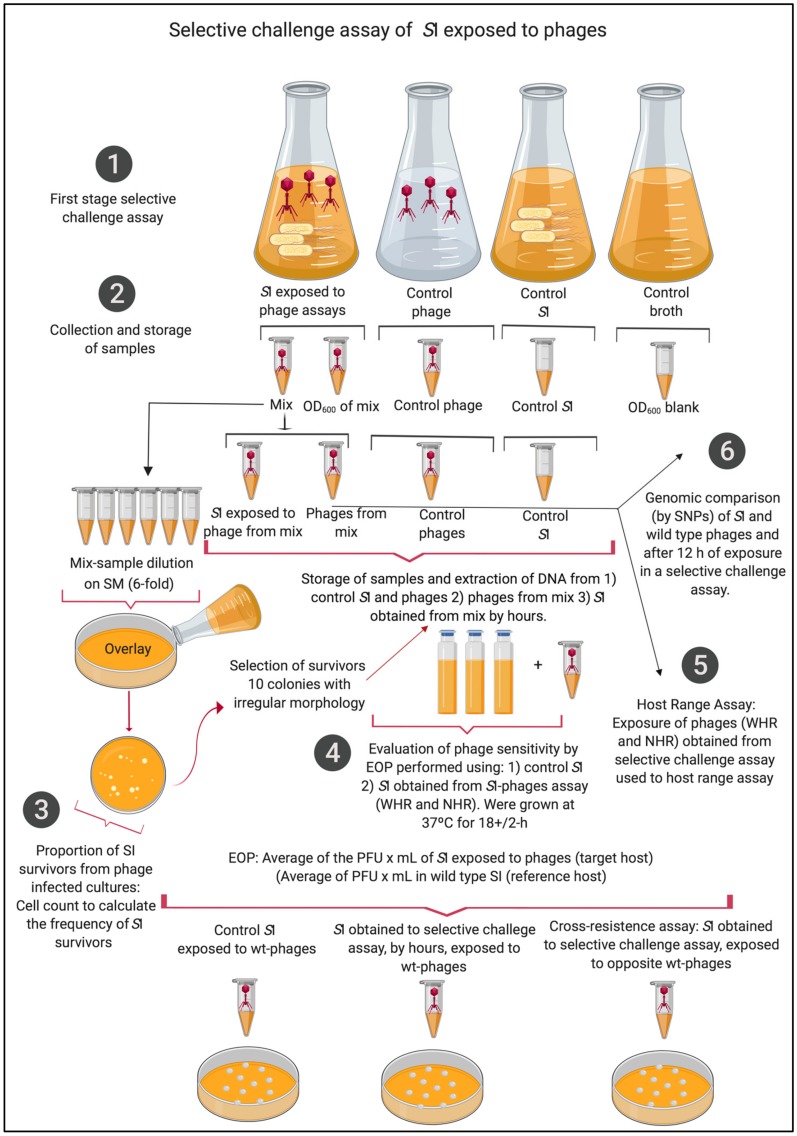
Selective challenge assays of *S.* Infantis exposed to wide- and narrow-host-range phages (WHR and NHR) at 12 h exposure. This assay was developed following the steps: (**1**) first stage selective challenge assay, (**2**) collection and storage of samples, (**3**) calculation of proportion of *S*I survivors from phage-infected cultures, (**4**) evaluation of phages susceptibility by efficiency of plating (EOP), (**5**) host-range assay, and (**6**) genomic comparison by nucleotide polymorphism (SNPs) of *S.* Infantis and phages at 12 h exposure.

**Figure 2 viruses-11-00586-f002:**
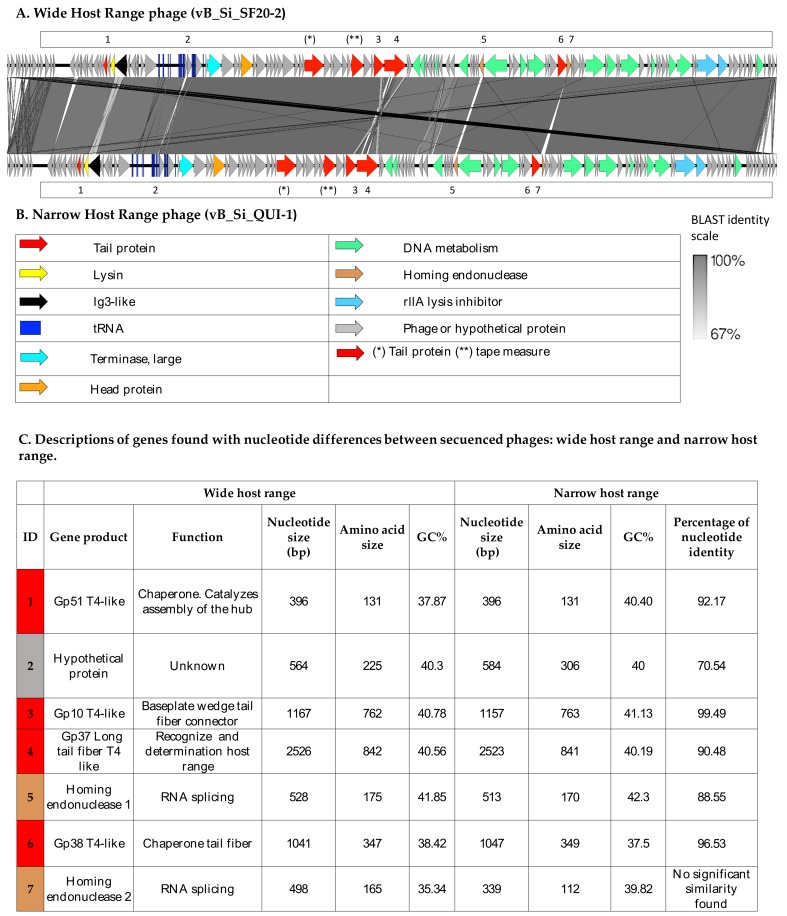
Linear representation of the genomes of wide- and narrow-host-range phages. EasyFig software version mc 2.1 [48] was used to construct a comparison using the BLAST algorithm of WHR and NHR phages. (**A**) Wide-host-range phage and (**B**) narrow-host-range phage. The presence of similar genes is shown in the figure. The genes were labeled with different colors according to functionality: tail proteins, lysin, Ig3-like, tRNA, terminase large, head protein, DNA metabolism, homing endonuclease, rlla lysis inhibitor. BLAST identity is represented in gray-scale, with dark gray indicating a greater BLAST identity and light gray representing a lesser BLAST identity. Numbers indicate genes with differences described in C; (**C**) description of genes found with differences in the shared nucleotide identity between both phages.

**Figure 3 viruses-11-00586-f003:**
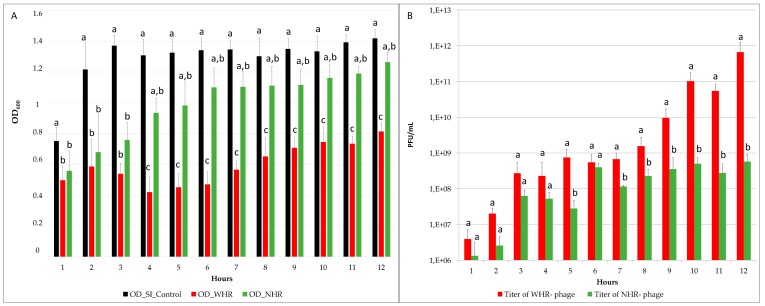
Graphical representation of parameters measured to compare challenge assays during the 12 h experiments. (**A**) Optical density (OD_600_) obtained per hour of control *S.* Infantis, *S.* Infantis exposed to WHR phages, and *S.* Infantis exposed to NHR phages. A statistical analysis was performed to compare the hourly differences between OD control, WHR, and NHR through Kruskal Wallis, *p* < 0.05). (**B**) Viral titer obtained per hour on challenge with WHR and NHR phages, expressed as PFU/mL. The differences in phage titer were analyzed through (ANOVA, *p* < 0.05). Standard deviation was calculated between the four replicates per hour, for each variable, and statistically significant differences (*p* < 0.05) are indicated by different letters.

**Figure 4 viruses-11-00586-f004:**
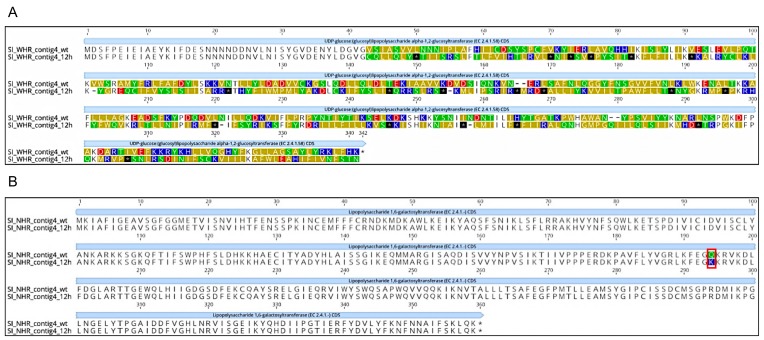
Amino acid alignment of putative receptor in *Salmonella* Infantis. (**A**) Amino acid alignment of UDP-D-glucose:lipopolysaccharide alpha-1,2-glucosyltransferase (UDP-D-1,2) in wild-type *S*. Infantis and *S*. Infantis exposed to WHR phage. (**B**) Amino acid alignment of UDP-D-galactose lipopolysaccharide galactosyltransferase 1,6 (UDP-D-1,6) in wild-type *S*. Infantis and *S.* Infantis exposed to NHR phage. The nucleotide sequence of the gene from the wild-type genome was extracted and modified to reflect the mutation, then the wild-type and mutation-containing nucleotide sequences were translated and aligned with ClustalW [75] in Geneious Prime [76]. Amino acids labeled with different colors represent differences between sequences of wild-type *S*I, and the 12 h exposed *S*I. Amino acid changes caused by single nucleotide polymorphisms (SNPs) are marked in red frames.

**Figure 5 viruses-11-00586-f005:**
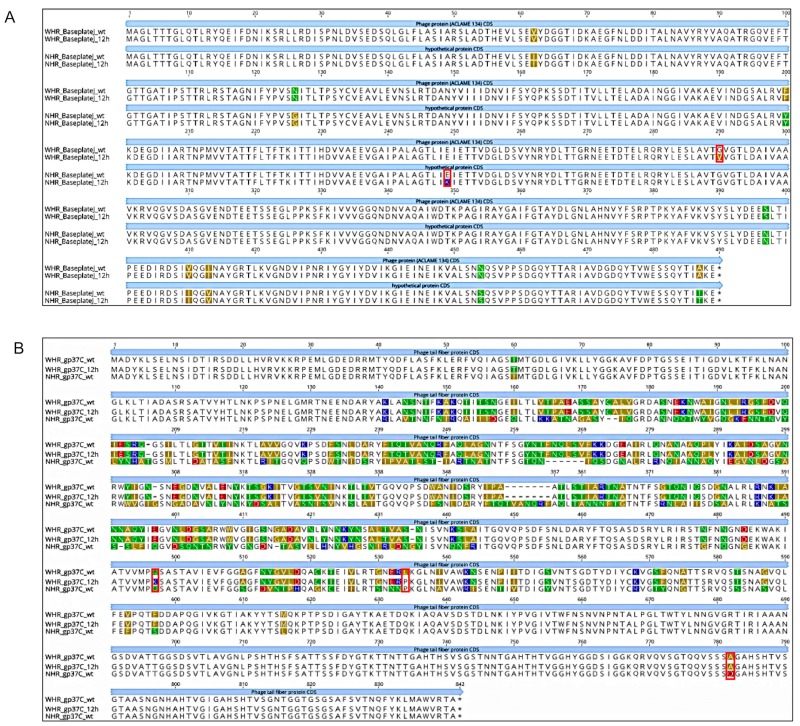
Amino acid alignment of baseplate J-like protein and gp37-like protein. (**A**) Amino acid alignment of baseplate J-like protein in wild-type WHR and NHR phages and WHR and NHR phages that had interacted with *S.* Infantis for 12 h. (**B**) Amino acid alignment of gp37-like protein in wild-type WHR and NHR phages and WHR phage exposed to *S.* Infantis for 12 h. The nucleotide sequences of the gene from the wild-type genomes were extracted and modified to reflect the mutation(s), then the wild-type and mutation-containing nucleotide sequences were translated and aligned with ClustalW [75] in Geneious Prime [76]. Amino acids labeled with different colors represent differences among sequences of wild-type phages and the 12 h-exposed phages. Amino acid changes caused by SNPs are marked in red frames.

**Table 1 viruses-11-00586-t001:** Genomic and phenotypic characteristics of wide host and narrow host range.

	Bacteriophages	
Characteristics	vB_Si_SF20-2	vB_Si_QUI-1
Host range classification	Wide host range (WHR)	Narrow host range (NHR)
*Salmonella* serovars lysed ^1^	Choleraesuis, Panama, Javiana, Kentucky, Montevideo, Infantis, Oranienburg, Typhimurium, Corvallis, Mbandaka, Dublin, Newport, Braenderup, Enteritidis	Kentucky, Infantis, Oranienburg, Typhimurium, Dublin, Braenderup
NCBI Accession Number	MK965970.1	MK965969.1
BLAST results with highest e-value of fully annotated phage (accession)	Mushroom (KP143762.1)	Felix 01 (AF320576.1)
Genome size of assembly	89,093 bp	89,218 bp
G + C content	39.76%	39.14%
Size of the capsid ^2^	73 × 69 nm	81 × 92 nm
Length of the tail	144 ± 3 nm	146 ± 3 nm
Burst size (viruses)	30 ± 5	12.6 ± 4
Latency time	40 ± 10 min	55 ± 15 min

^1^ Previously reported [40]; ^2^ Approximate value; three measurements of the same image were obtained, and the average is reported.

**Table 2 viruses-11-00586-t002:** Proportion of *S.* Infantis survivors after exposure to wide- and narrow-host-range phages in the selective challenge assays.

Hours	Average ^1^ of Proportions of *S*I Survivors Exposed to WHR ^2^ Phage	Average of Proportions of *S*I Survivors Exposed to NHR ^3^ Phage
1	8.1 × 10^−6^	8.8 × 10^−6^
2	4.2 × 10^−6^	1.0 × 10^−5^
3	8.8 × 10^−6^	4.0 × 10^−5^
4	8.8 × 10^−7^	3.1 × 10^−6^
5	3.3 × 10^−7^	2.0 × 10^−6^
6	2.1 × 10^−6^	7.7 × 10^−6^
7	3.5 × 10^−6^	1.2 × 10^−5^
8	7.6 × 10^−7^	1.5 × 10^−6^
9	8.5 × 10^−7^	1.2 × 10^−6^
10	1.2 × 10^−6^	1.8 × 10^−6^
11	8.1 × 10^−6^	1.2 × 10^−6^
12	4.2 × 10^−6^	9.7 × 10^−7^
**Average**	2.7 × 10^−6^	7.6 × 10^−6^

^1^ Average of four replicates; ^2^ Wide-Host-Range phage; ^3^ Narrow-Host-Range phage.

**Table 3 viruses-11-00586-t003:** Identification of variants in *S.* Infantis exposed to phage and in phages obtained from selective challenge assay after 12 h exposures.

Variant Information				*S*I Unexposed to WHR Phage (control)			*S*I Exposed to WHR Phage 12 h			
CDD and pfam ID ^1^	Genome Position (contig)	Reference Allele	Alternative Allele	Kmer Coverage of Reference Allele	Kmer Coverage of Alternative Allele	Variant Frequency (%)	Kmer Coverage of Reference Allele	Kmer Coverage of Alternative Allele	Variant Frequency (%)	Protein Effect ^7^
UDP-D-1,2 ^2^ (PRK10124)	94397 (4)	TG	T	63	0	0.00	0	109	100.00	Truncated protein ^3^
**Variant Information**				***S*I unexposed to NHR phage (control**)			***S*I exposed to NHR phage 12 Hour**			
UDP-D-1,6 ^4^ (PRK09922)	92739 (4)	C	A	92	0	0.00	0	64	100.00	Radical Nonsynonymous Substitution
**Variant Information of WHR**				**WHR phage (Control**)			**WHR phage 12 Hour**			
Baseplate J-like protein (pfam04865)	17509 (1)	G	T	42	10	19.23	11	1	8.33	Conservative Nonsynonymous Substitution
(FNI) ^5^_Phage protein (a)	18756 (1)	C	A	7	29	80.56	0	2	100.00	Radical Nonsynonymous Substitution
(FNI) ^5^_Phage protein (b)	18756 (1)	C	T	7	24	77.42	0	3	100.00	Conservative Nonsynonymous Substitution
gp37-like ^6^ (pfam12604)	21980 (1)	C	A	27	30	52.63	0	5	100.00	Radical Nonsynonymous Substitution
gp37-like ^6^ (pfam12604)	22092 (1)	T	C	21	32	60.38	0	6	100.00	Conservative Nonsynonymous Substitution
**Variant Information of NHR**				**NHR phage (Control**)			**NHR phage 12 Hour**			
Baseplate J-like protein (pfam04865)	17384 (1)	G	A	7	8	53.33	5	17	77.27	Radical Nonsynonymous Substitution
(NI)_Hypothetical protein	40239 (1)	T	G	3	1	25.00	13	0	0.00	Radical Nonsynonymous Substitution

^1^ Conserved Domains Database (CDD) is a protein annotation resource that consists of a collection of well-annotated multiple sequence alignment models for ancient domains and full-length proteins, and Pfam is an extensive collection of multiple alignments of Markov’s hidden sequences and models covering much of the protein domains and common families; ^2^ Corresponding to gene *rfaK* that encodes UDP-D-glucose:lipopolysaccharide alpha-1,2-glucosyltransferase (UDP-D-1,2); ^3^ Frameshift (causing premature stop codon downstream); ^4^ Corresponding to gene *rfaB* that encodes UDP-D-galactose lipopolysaccharide 1,6-galactosyltransferase (UDP-D-1,6); ^5^ Functionality not identified (FNI); ^6^ long tail fiber gp37-like; ^7^ Nonsynonymous substitutions were classified as conservative or radical based on amino acid charge and polarity (Hanada et al.) [77]. The amino acid alignments of the phage protein (a and b) in WHR and Hypothetical protein in NHR are shown in Appendix A.

**Table 4 viruses-11-00586-t004:** Phenotypic assay of efficiency of plating on *S.* Infantis obtained at 12 h of exposure to wide- and narrow-host-range phages in selective challenge assays.

	*S.* Infantis Obtained from Exposure to Wide-Host-Range (WHR) Phage					*S.* Infantis Obtained from Exposure to Narrow-Host-Range (NHR) Phage				
	EOP ^1^ of Each Replicate (Replicates)					EOP ^1^ of Each Replicate (Replicates)				
Phage	(R1)	(R2)	(R3)	(R4)	Average of replicates	(R1)	(R2)	(R3)	(R4)	Average of replicates
Wild-type WHR phage	0.2	1	0	0.1	0.3	0	0	0	0	0
Wild-type NHR phage	1	0.1	0	1	0.5	0	0	0	0	0

^1^ EOP (efficiency of plating) was calculated as described previously [54], see methods.

**Table 5 viruses-11-00586-t005:** Differences in the host range in the four replicates of wide- and narrow-host-range phages obtained at 12 h of the selective challenge assay.

Host range assay of wide-host-range phage obtained from each replicate (Nº of a given replicate)						Host range of narrow-host-range phage obtained from each replicate (Nº of a given replicate)				
	Wild-type	R1	R2	R3	R4	Wild-type	R1	R2	R3	R4
Serovars lysed *	CHO PAN JAV KEN MON INF ** ORA TYP COR MBA DUB NEW BRA ENT	MUE VIR SAI CHO PAN JAV KEN MON INF ** ORA TYP AGO HEI NEW STA 4,5,12:i:-	VIR, SAI CHO PAN JAV MON INF ** ORA TYP AGO HEI NEW 4,5,12:i:-	CHO MON INF ** ORA	CHO MON INF ** BRA	KEN INF ** ORA TYP DUB BRA	ORA NEW	ORA TYP PAN COR NEW	VIR JAV ORA NEW	ORA NEW
Total serovars lysed	14	16	12	4	4	6	2	5	4	2

* Serovar abbreviations: VIR, Virchow; SAI, Saintpaul; CHO, Choleraesuis; PAN, Panama; JAV, Javiana; KEN, Kentucky; MON, Montevideo; INF, Infantis; ORA, Oranienburg; TYP, Typhimurium; AGO, Agona; COR, Corvallis; MBA, Mbandaka; DUB, Dublin; NEW, Newport; BRA, Braenderup; ENT, Enteritidis; MUE, Muenster; HEI, Heidelberg; STA, Stanley; WEL, Weltevedren. ** INF, Infantis corresponds to the wild-type host of isolation.

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
