# Peer review of "Two Phages of the Genera Felixunavirus Subjected to 12 Hour Challenge on Salmonella Infantis Showed Distinct Genotypic and Phenotypic Changes"

_viruses, 2019, doi:10.3390/v11070586_

Reviewer 1 Report

The review of Rivera et al. refers to the characterization of two bacteriophages of the Felixounavirus genus and specific to Salmonella enterica serovar Infantis. In addition, they identify, by DNA sequencing of S. Infantis isolates, changes in some genes that could be the cause of the resistance to these phages. Furthermore, they describe changes in some proteins of these bacteriophages obtained from these cultures.

Broad comments

The introduction is appropriate and underlines some aspects related to bacteriophage resistance and Felixounavirus genus characteristics needed to justify the objectives of the study.

Material and methods section is well written, although information about the production of bacteriophage lysates and the S. Infantis production strain could be included. Similarly, the number of isolates obtained in the different studies of EOP and the SI survivors must be indicated. In addition, the final number of SI survivors sequenced and which was the criteria for selecting those that were sequenced. This could be indicated in the material and methods or results section as it is of interest to know the number of survivors and which of them were sequenced. Similarly, for those bacteriophages selected for sequencing out of those recovered after infection studies.

Concerning the results and discussion section, some points need to be explained and discussed in a higher extent. For example, differences in the genomes of the two phages represented in Figure 2. Except in some cases as a homing endonuclease that seems is moved in position through the genomes of both phages, it is difficult to ascertain the differences with respect the rest of genes mentioned in the text. In other sense, the high standard error observed in the WHR phage titer represented in Figure 3 must be discussed or revised. How do the authors explain these results? Depending on this, the production of phages could be considered as not relevant. Reproducibility of the assays must be revised.

In the other hand, it would be of interest that SNPs results could be confirmed by complementation of affected genes and new infection studies. This would reinforce the results described. In addition, it is difficult to extract what it the importance that SNPs detected have with respect to the objectives of the study as only the results are exposed but scarcely discussed. Which is the implication of the SNPs regarding the developing of phage resistance? Do these variants confer better capacity of infection to phages? If this, how is explained that certain variants appear with higher frequency in a non-exposed phage than in one 12h-exposed?

Furthermore, although the authors declare that WHR does not induce specific resistance or cross-resistance, the reduction of EOP indicates that something happens upon phage infection (i.e. by restriction activation or delaying of DNA replication, etc). Authors indicate that phage modifies itself but without setting the cause.

For example, if NHR phage produces specific and cross-resistance after 12h of exposition, how is explained the increase (although lower than WHR) in host range for some NHR phage isolates?

Minor comments

Line 33. Please, rewrite the superscript of the titer of WHR phage.

Line 78: Please, rewrite as Caudovirales.

Table 1. Please, add the reference of your previous study about data of the serotypes lysed by the phages isolated. Also, some serotypes lysed by vB_Si_SF20-2 are missing in this table comparing to the results exposed in Rivera et al., 2018. Similarly, serotypes exposed for vB_Si_QUI-1 VIR, INF, ORA, and NEW are not the same referred in the mentioned study (KEN, INF, ORA, TYP, DUB, BRA). Please, revise. Similarly, for data exposed in Table 5.

Lines 197: How did the authors confirm that phages were eliminated? This must be clarified.

Line 200: This sentence needs clarification as it seems there is a mistake. In the literature, EOP values of 1 are considered as high efficiency and similar to wild type strains. Those values reaching to 0 evidence very low efficiency. Confirm this with the reference you indicated (Petsong et al., 2019). Moreover, this is in contrast to the exposed in table 4 and the results and discussion section (3.4).

Line 218. “SI” is duplicated.

Lines 270, 272, 503, and 505. Please, revise if it is correct. The most adequate term would be plaques.

Line 286: Please, change “tRNa” by “tRNA”.

The software used for genome viewing on figure 2 must be indicated in the figure legend or material and methods.

Line 339: eliminate the first a in “(a; a, b, and c)”.

Line 345: Change “6.6 x 10-6” by “7.6 x 10-6” similar to Table 2 data.

Line 362: Please, revise the sentence “to obtain the SNPs of importance”.

Lines 375 to 377 and lines 377 to 379: It seems both paragraphs explain the same using different references. Please, revise or rewrite.

Table 3: (FNI)5 variant position number is the same in two files of the table. Please, rewrite. If the information comes from the same genome, it is not possible to have different changes at the same sequence position.

Line 398. Does “gene phage protein” refer to Hypothetical protein in Table 3 for NHR phage?

Line 437: With respect to the data exposed in table 4, the value of “0.2” is a mistake. Please, change.

Lien 474: Delete “as”.

Line 474. It is not “three”, it is “four” for R3 and “four” and not “five” for R4. This last has to be corrected in table 5.

Table 5. Correct “R2” by “R1” in narrow host range first column.

References: Please, revise the reference list and eliminate points (i.e. line 554, 612, 617, 658, 747, and 757) also some years are missing, etc. In addition, the style of references must be standardized according to the format of the journal Viruses.

Author Response

Response to Reviewer 1 Comments

Comment 1: The introduction is appropriate and underlines some aspects related to bacteriophage resistance and Felixounavirus genus characteristics needed to justify the objectives of the study.

Response 1. We acknowledge this comment and have made minor revisions to the introduction

Comment 2. Material and methods section is well written, although information about the production of bacteriophage lysates and the S. Infantis production strain could be included. Similarly, the number of isolates obtained in the different studies of EOP and the SI survivors must be indicated. In addition, the final number of SI survivors sequenced and which was the criteria for selecting those that were sequenced. This could be indicated in the material and methods or results section as it is of interest to know the number of survivors and which of them were sequenced. Similarly, for those bacteriophages selected for sequencing out of those recovered after infection studies.

Response 2. We have clarified in methods these comments. Specifically, we have added the methods used for the production of SI on lines 108 and 113-117 for the production of phage lysates. The number of survivors isolated was added to line 167-173. We added the number of isolates used for the EOP assays in line 200-209.

On lines 221 and 231 we added the criteria to select the isolates and phages for sequencing and the number of isolates sequenced.

Comment 3: Concerning the results and discussion section, some points need to be explained and discussed in a higher extent. For example, differences in the genomes of the two phages represented in Figure 2. Except in some cases as a homing endonuclease that seems is moved in position through the genomes of both phages, it is difficult to ascertain the differences with respect the rest of genes mentioned in the text.

Response 3. To clarify differences in the two phage genomes, we have added a detailed comparison on the regions that presented differences in Figure 1. This includes the percentage of nucleotide identity in each of the genes that presented differences.

Comment 4: In other sense, the high standard error observed in the WHR phage titer represented in Figure 3 must be discussed or revised. How do the authors explain these results? Depending on this, the production of phages could be considered as not relevant. Reproducibility of the assays must be revised.

Response 4. Figure 3 was modified and the standard deviation between replicates was calculated as opposed to the old graphs that calculated the standard deviation between WHR and NHR test systems.

Comment 5: In the other hand, it would be of interest that SNPs results could be confirmed by complementation of affected genes and new infection studies. This would reinforce the results described. In addition, it is difficult to extract what it the importance that SNPs detected have with respect to the objectives of the study as only the results are exposed but scarcely discussed. Which is the implication of the SNPs regarding the developing of phage resistance? Do these variants confer better capacity of infection to phages? If this, how is explained that certain variants appear with higher frequency in a non-exposed phage than in one 12h-exposed?

Response 5. Genetic complementation is great for providing strong evidence for a mechanism. However, the goal of the present study was to determine what mutations occur, not to confirm the mechanism of the interaction. This would be a great follow-up study to perform. We have presented kmer coverage and frequency for each of the SNP positions and the position of these mutations within certain gene coding regions make logical sense based on what we know about phage receptors.

We added more information into discussion about the roles of the genes that contain mutations (in the context of phage infection) and how these mutations could affect phage resistance in SI or overcoming resistance in the phages, on lines 417-443 and Figure 4.

Additionally, we have added information about phage tail fiber and baseplate proteins, their role in adsorption, and how modifications in genes encoding these proteins could aid phage in overcoming resistant hosts, on lines 463-485 and Figure 5.

Comment 6. Furthermore, although the authors declare that WHR does not induce specific resistance or cross-resistance, the reduction of EOP indicates that something happens upon phage infection (i.e. by restriction activation or delaying of DNA replication, etc). Authors indicate that phage modifies itself but without setting the cause

Response 6. We added to lines 552-572 and 587-591 some discussion on hypothesis related with cross-resistance.

Comment 7. For example, if NHR phage produces specific and cross-resistance after 12h of exposition, how is explained the increase (although lower than WHR) in host range for some NHR phage isolates?

Response 7.  We have revised errors in Table 5 and Table 1. These tables were revised.  Table 5 now shows that for NHR phage, the host range decerased in all replicates in comparison with the wildtype phage.  We added on lines 546-563 and 597-601, more details describing the fact that NHR phage lost the ability lyse to wild type S. Infantis.

Comment 8. Line 33. Please, rewrite the superscript of the titer of WHR phage.

Response 8. Change was made

Comment 9. Line 78: Please, rewrite as Caudovirales.

Response 9. Was corrected on line 78

Comment 10. Table 1. Please, add the reference of your previous study about data of the serotypes lysed by the phages isolated. Also, some serotypes lysed by vB_Si_SF20-2 are missing in this table comparing to the results exposed in Rivera et al., 2018. Similarly, serotypes exposed for vB_Si_QUI-1 VIR, INF, ORA, and NEW are not the same referred in the mentioned study (KEN, INF, ORA, TYP, DUB, BRA). Please, revise. Similarly, for data exposed in Table 5.

Response 10. A footnote was added to Table 1 that states “Previously reported” with a reference to the previous study. It was checked and there was an error that was revised in Table 1 and Table 5.

Comment 11. How did the authors confirm that phages were eliminated? This must be clarified

Response 11. This was corrected on line 202. Isolates obtained from each of the four replicates of the challenges were successively inoculated on TSA for two passages, until, verifying the absence of phage plaques.

Comment 12. This sentence needs clarification as it seems there is a mistake. In the literature, EOP values of 1 are considered as high efficiency and similar to wild type strains. Those values reaching to 0 evidence very low efficiency. Confirm this with the reference you indicated (Petsong et al., 2019). Moreover, this is in contrast to the exposed in table 4 and the results and discussion section (3.4).

Response 12. We have a clarification on lines 205-206 that stated “results were expressed on a scale of complete resistance (0) to complete sensitivity (1), as previously reported [54]

Comment 13.  Line 218. “SI” is duplicated.

Response 13: Was corrected on line 213

Comment 14.  Lines 270, 272, 503, and 505. Please, revise if it is correct. The most adequate term would be plaques.

Response 14. Revision was added on line 293-294, 677 and 678

Comment 15.  Line 286: Please, change “tRNa” by “tRNA”.

The software used for genome viewing on figure 2 must be indicated in the figure legend or material and methods.

Response 15. This was corrected on lines 317 and 1067 and in the figure caption.

Comment 16. Eliminate the first a in “(a; a, b, and c)”.

Response 16. It was eliminated in the figure legend

Comment 17. Change “6.6 x 10-6” by “7.6 x 10-6” similar to Table 2 data.

Response 17.  Change was made on lines 381-382.

Comment 18. Line 362: Please, revise the sentence “to obtain the SNPs of importance”.

Response 18. Sentence was edited on lines 249-259

Comment 19: Lines 375 to 377 and lines 377 to 379: It seems both paragraphs explain the same using different references. Please, revise or rewrite.

Response 19. These sentences were rewritten.

Comment 20: Table 3 (FNI) 5 variant position number is the same in two files of the table. Please, rewrite. If the information comes from the same genome, it is not possible to have different changes at the same sequence position.

Response 20: Actually, this is possible because the phage sample sequenced likely contain subpopulations (as opposed to the genome of a single phage). At this nucleotide position, two different mutations were observed: C -> A and C -> T.  We have updated the manuscript text to more clearly explain that samples sequenced contained subpopulations, on line 410-416 and 470-476.

Comment 21: Line 398. Does “gene phage protein” refer to Hypothetical protein in Table 3 for NHR phage?

Response 21. We revised the text to clarify that

No, it refers to the gene annotated as “hypothetical protein” in WHR phage. This has been clarified in the text.

Comment 22: Line 437: With respect to the data exposed in table 4, the value of “0.2” is a mistake. Please, change.

Response 22: Was corrected in Table 4.

Comment 23. Line 474 Delete “as”.

Response 23. Was corrected

Comment 24: Line 474. It is not “three”, it is “four” for R3 and “four” and not “five” for R4. This last has to be corrected in table 5.

Response 24: Changes were made on lines 589-590 and Table 5.

Comment 25: Table 5. Correct “R2” by “R1” in narrow host range first column.

Response 25. It was corrected in table 5

Comment 26: References: Please, revise the reference list and eliminate points (i.e. line 554, 612, 617, 658, 747, and 757) also some years are missing, etc. In addition, the style of references must be standardized according to the format of the journal Viruses.

Response 26. We revised the reference list in all the manuscript.

Reviewer 2 Report

The authors present data on a coevolution of bacteriophages and host cells. The presented results might be important and applied for improvement of the existing phage-based therapies. However, the manuscript cannot be accepted for publishing in its current form, since the paper suffers from a lack of attention to detail, both in terms of the presentation of the material and the essential analysis. The paper has numerous grammar and language issues (only a few of many are listed below), which need to be addressed.

Questions, comments and suggestions.

Keeping in mind the different host spectra of the phages, the description and a comparative analysis of the putative receptors should be included especially in the light of selected mutants of the LPS synthesis. Any detailed discussion regarding the host recognition would be welcomed. Moreover, the changes in the host range presented in the paragraph “3.5 Phenotypic modification of host range in WHR and NHR phages after 12 hour assays” should be also discussed in more details.

The functions of gp37-like proteins are well known and studied. It is not unexpected that gp37 mutants are selected in the case of WHR phage, but why analogous mutations are not observed in the NHR phage? It would be also interesting and informative to the readers if the locations of the selected mutations are presented on the model of the gp37 from WHR. The comparison/alignment of gp37s from both phages might be included and discussed. A hypothesis regarding the differences in accumulated mutants of LPS biosynthesis would be also welcomed.  

The names of bacteria as well as Latin words such as “in vitro” should be in italic throughout the text.

Lines 29, 31 The text “SI OD600nm” and “Salmonella OD decreased” should be corrected since it is misleading

Lines 59, 60 Reiterations “gaps in knowledge” should be avoided.

Table 1 “+/-“ may be changed to “±”. What does the value “73x69 nm (+/-)” as well as “40 min(+/-10)” mean?

Author Response

Response to Reviewer 2

Comment 1. Keeping in mind the different host spectra of the phages, the description and a comparative analysis of the putative receptors should be included especially in the light of selected mutants of the LPS synthesis. Any detailed discussion regarding the host recognition would be welcomed.

Response 1. We added more information into discussion about putative phage receptors in Salmonella and the roles of the genes that contain mutations (in the context of phage infection) and how these mutations could affect phage resistance in SI. (lines 417-443 and 612-637) and Figure 4

Comment 2. Moreover, the changes in the host range presented in the paragraph “3.5 Phenotypic modification of host range in WHR and NHR phages after 12 hour assays” should be also discussed in more details.

Response 2. We have added a discussion on the variability found between replicates on line 586-601.

Comment 3. The functions of gp37-like proteins are well known and studied. It is not unexpected that gp37 mutants are selected in the case of WHR phage, but why analogous mutations are not observed in the NHR phage? It would be also interesting and informative to the readers if the locations of the selected mutations are presented on the model of the gp37 from WHR. The comparison/alignment of gp37s from both phages might be included and discussed. A hypothesis regarding the differences in accumulated mutants of LPS biosynthesis would be also welcomed. 

Response 3.

We have included new figures, as Figure 5 that represents an alignment of gp37 for NHR and WHR phages. In addition,  as supplemental figures (S2) an alignment of other proteins listed in Table 3. Additionally, we have added more explanation of the role of enzymes in LPS synthesis to the results/discussion section (section 3.3).

Comment 4. The names of bacteria as well as Latin words such as “in vitro” should be in italic throughout the text.

Response 4. This was modified throughout the document

Comment 5. Line 29, 31 The text “SI OD600nm”

Response 5. This was modified on line 31

Comment 6. and “Salmonella OD decreased” should be corrected since it is misleading ”

Response 6. It was modified on lines 31-33

Comment 7. Lines 59, 60 Reiterations “gaps in knowledge” should be avoided.  

Response 7. It was modified on line 59-62

Comment 8. Table 1 “+/-“ may be changed to “±”. What does the value “73x69 nm (+/-)” as well as “40 min(+/-10)” mean?

Response 8. It was modified on Table 1

Reviewer 3 Report

The manuscript by D. Rivera et al described the results of genotypic and phenotypic changes after S. infantis infected 2 phages for 12 h. The results obtained are quite interesting and showed that both bacteria and phages were mutated rapidly during the process. These data are important for understanding the interactions between phages and bacteria. The manuscript could be improved by providing more details.

1.      For the part 3.5 phenotypic modification of host range in WHR and NHR phages, how were the replicates of the phages obtained? Why are the host ranges varied so much for different replicates?

2.      In Table 5, there are some spelling errors, “reply” should be “replicate” and there are two “R2” for NHR phage.

3.      For the part 3.3 when analyzing the genotypic responses, could the method identify the existence of or ratios between different genotypic types of phages or bacteria in the solutions after the 12 h challenge? We could image that there would exist phages with different mutations in their genomes after 12 h growth/interaction. By extracting the DNA from all the phages in the solution, the sequencing will get the meta-genomes of  all the phages, which might include some reads from host genome also. Would the analysis methods used in the study differentiate the phages with few mutations in their genomes?

4.      The phages are DNA viruses. Normally people believe that DNA viruses are relatively stable during replication, which is quite different from the current results. Could the author explain the mutations in the genomes of either the bacteria or phages are induced due to the phage-bacteria interaction or simply due to the random mutations during DNA replication or sequencing errors, especially since SNPs were used in the analysis?

5.      With the four main findings in this study, could the authors add some discussion to state how these findings would impact on overcoming phage resistance in phage therapy? A mission impossible?

Author Response

Response to Reviewer 3 Comments

Comment 1. For the part 3.5 phenotypic modification of host range in WHR and NHR phages, how were the replicates of the phages obtained? Why are the host ranges varied so much for different replicates?

Response 1. We added to the methods how the replicates were conducted (lines 221-232).

It is possible that it has produced an important variation between replicates, because we work with biological replicates, which implies that each assay was carried out at different times, under the same, experimental conditions, from different stocks of phages obtained from a single lysis plate and different cultures of SI obtained from the same initial bacterial strain. Each stock of phage and bacterial culture was analysed by PCR, to ensure its suitability on lines (235-238).

It is very interesting to mention that this variation was more important for WHR phages than for NHR and was observed preferably in the host range and in the susceptibility analysis of SI by EOP. This difference was not observed in other measured parameters such as titre, neither in the proportion of survivors.

Comment 2. In Table 5, there are some spelling errors, “reply” should be “replicate” and there are two “R2” for NHR phage. R: in table 5 line 503

Response 2.  It was modified in Table 5 and throughout the document.

Comment 3. For the part 3.3 when analysing the genotypic responses, could the method identify the existence of or ratios between different genotypic types of phages or bacteria in the solutions after the 12 h challenge? We could image that there would exist phages with different mutations in their genomes after 12 h growth/interaction.

Response 3. Effectively can be identified the answer are mentioned on the lines 400-416

According to the selection criteria for useful mutations mentioned above, we compared them to the SNPs in the SI control strain and also phages (WHR and NHR). This consideration allowed us to filter out all spontaneous mutations given by the time of exposure, as previously described [18,71]. Through variant analysis, it was possible to identify subpopulations of genes Phage protein and gp37-like in WHR phages as opposed to those observed for SI strains exposed to WHR and NHR phages for which gene variants were not observed in SNPs filtered by the selection criteria of useful SNPs.

In Table 3, we have included frequencies for the mutations in each sample (kmer coverage of the reference allele / kmer coverage of the alternative allele). This should be representative of the phage population present at 12 h. However, individual phage genomes were not differentiated.

Comment 4. By extracting the DNA from all the phages in the solution, the sequencing will get the meta-genomes of  all the phages, which might include some reads from host genome also.

Response 4. It is unlikely because the protocol to separate phages from host includes in the case of phages: the filtration of stock and the incorporation of chloroform and in the case of bacteria, before sending to sequence was verified that they did not have plaques of lysis. Additionally, the DNA extraction protocol incorporates poteinase K and other reagents that could damage the integrity of phages. To send to sequence were separated phage and SI samples as explained in the selective challenge. Previously to send to sequence the stock of phage and the strains of SI collected were verified. The DNA extracted from SI was analyzed by PCR to verify the absence of DNA from WHR and NHR phages, using the specific genes mentioned above. In reverse for phages, the absence of DNA from SI was verified using the PCR from the invA gene mentioned above. Is clarified on 183-189 lines and 223-235 of materials and methods.

Comment 5. Would the analysis methods used in the study differentiate the phages with few mutations in their genomes?

Response 5. While the methods used can always been improved, our methodology worked to identify from 2 mutations in NHR phage to five in WHR phage.

Comment 6. The phages are DNA viruses. Normally people believe that DNA viruses are relatively stable during replication, which is quite different from the current results. Could the author explain the mutations in the genomes of either the bacteria or phages are induced due to the phage-bacteria interaction or simply due to the random mutations during DNA replication or sequencing errors, especially since SNPs were used in the analysis?

Response 6. All mutations that are reported in this manuscript should be random mutations that occurred during DNA replication. However, we only reported mutations that were enriched in the experimental flasks (i.e., mutations only found in the phage-infected SI, or mutations only found in the phages that were coevolved on SI). We also report i) the Kmer Coverage of reference allele and alternative allele for each gene ii) percentage of the variant in the reference and alternative allele and iii) approximate probability of occurrence of the variant in reference and alternative allele (Variant P-Value ). On lines 240-259 and Table 3. The coverage of reported mutations with high quality Illumina reads make sequencing errors passing our selection criteria very unlikely.

Comment 7. With the four main findings in this study, could the authors add some discussion to state how these findings would impact on overcoming phage resistance in phage therapy? A mission impossible?

Response 7

Although the results obtained represent the interaction of two WHR and NHR phages against a single strain of Salmonella Infantis, these results show the great importance of develop challenges with its host, to later evaluate the viral titre, phages resistant mutants, host range, genetic changes. All factors necessary to design rational and effective biocontrol applications based on phages  (lines 650-661).

Round  2

Reviewer 1 Report

The manuscript has been revised and rewritten. However, some paragraphs (i.e. lines 612 to 637 and 650 to 661) need further revision and rewrite in order to understand the reasoning of the authors about their highlighted results.

Some of the results are hypothesized and would require confirmation although it seems that it is not the focus of this manuscript.

Minor comments

Line 180: Please, change "plaquing" by plating.

Lines 597 to 601: Perhaps it is a misunderstanding. Table 5 results expose the capacity of replicates of WHR and NHR phages to infect different serovars of Salmonella. Figure S3 and S4 show the incapacity of NHR wt phage to infect SI replicates exposed to WHR or NHR. In the first case, it is supposed that NHR phage replicates (R1 to R4) suffer some changes that prevent the infection of wt SI strain. In the second, SI replicates modify something to avoid the infection by the wt NHR phage. Please, clarify.

Figure S2 (A): What is the meaning of WHRa and b? variants of the same gene reflected in Table 3?

The quality of some images (i.e. Figure 5) is low. For this reason, it is difficult to ascertain for example, why in Baseplate gene comparison (wt vs 12) are highlighted amino acids that seem the same, besides those different. Please, revise.

Author Response

Reviewer 1

Comment 1. The manuscript has been revised and rewritten. However, some paragraphs (i.e. lines 612 to 637 and 650 to 661) need further revision and rewrite in order to understand the reasoning of the authors about their highlighted results. Some of the results are hypothesized and would require confirmation although it seems that it is not the focus of this manuscript.

Response comment 1. It was clarified on lines 621-626 and 636-647. Main changes include to report that further confirmation is required and we have also shortened this section.

Comment 2. Line 180: Please, change "plaquing" by plating.

Response comment 2. It was modified throughout the document.

Comment 3. Lines 597 to 601: Perhaps it is a misunderstanding. Table 5 results expose the capacity of replicates of WHR and NHR phages to infect different serovars of Salmonella. Figure S3 and S4 show the incapacity of NHR wt phage to infect SI replicates exposed to WHR or NHR. In the first case, it is supposed that NHR phage replicates (R1 to R4) suffer some changes that prevent the infection of wt SI strain. In the second, SI replicates modify something to avoid the infection by the wt NHR phage. Please, clarify.

Response comment 3. We have clarified that SI exposed to NHR phage may have changed to avoid infection with wt NHR phage (line 681-693).

Comment 4. Figure S2 (A): What is the meaning of WHRa and b? variants of the same gene reflected in Table 3?

Response comment 4. It was modified on lines 469-470, in Table 3, and in Figure S2 caption to indicate what are the variants

Comment 5. The quality of some images (i.e. Figure 5) is low. For this reason, it is difficult to ascertain for example, why in Baseplate gene comparison (wt vs 12) are highlighted amino acids that seem the same, besides those different. Please, revise.

Response comment 5. We have submitted figures with better resolution.

In all of the amino acid alignment figures, the amino acids that are different between the different genomes being compared are highlighted different colors. So for some of them, this is in just one place (i.e. where the mutation occurs, like in the phage protein alignment figure). In others, there are multiple places where the amino acids are highlighted different colors (i.e., the gp37 alignment figure, as the sequences for the mutant and both wild-types are being compared; so there are multiple places where the amino acids differ).

Specifically in the baseplateJ alignment figure, the amino acids that are different between the WHR and NHR phages are highlighted. The amino acids might be the same between the WHR wt and the WHR 12h, but they are different than the NHR amino acids in that position. So these highlighted amino acid differences are not mutations, but just differences between the amino acid sequences of the WHR and NHR phages (which one reviewer suggested we show during the first review).

In addition, amino acids that were changed by mutations were marked in the figures (in red frames), so that they can be easily distinguished.  The legend of this figure was also modified to clarify this.

Reviewer 2 Report

The authors appropriately addressed most of the raised questions and improved the manuscript adequately.

Some minor comments and suggestions.

The mutations in Figures should be presented more clearly and should be easily recognizable, for example framed or marked by arrows, etc.

Regarding mutations of gp37-like protein it would be welcomed to include a sentence explaining the putative role of those amino acid positions in the recognition of receptors, since it looks like that the mutated region is not located in the receptor binding domain, which is located in the C-terminal part of protein (Bartual et al., Proc Natl Acad Sci U S A. 2010 107(47): 20287–20292; this reference should be also included). Moreover, keeping in mind very different host spectra, a single alteration (position 782) in this domain between two analysed phages is amazing. It would be really interesting to mutate this amino acid and test if the range of the hosts would be changed, too. Definitely, such experiment is out of scope of this work.

Author Response

Reviewer 2

Comment 1. The mutations in Figures should be presented more clearly and should be easily recognizable, for example framed or marked by arrows, etc.

Response comment 1. In all of the alignment figures, amino acids that were changed by mutations were marked in the figures (in red frames), so that they can be easily distinguished.

Comment 2. Regarding mutations of gp37-like protein it would be welcomed to include a sentence explaining the putative role of those amino acid positions in the recognition of receptors, since it looks like that the mutated region is not located in the receptor binding domain, which is located in the C-terminal part of protein (Bartual et al., Proc Natl Acad Sci U S A. 2010 107(47): 20287–20292; this reference should be also included). Moreover, keeping in mind very different host spectra, a single alteration (position 782) in this domain between two analysed phages is amazing. It would be really interesting to mutate this amino acid and test if the range of the hosts would be changed, too. Definitely, such experiment is out of scope of this work.

Response comment 2. The recommendation is appreciated.  We have marked the SNPs identified in Figure 5. We added the reference and a discussion on the putative role (lines 516-525)

Viruses EISSN 1999-4915 Published by MDPI AG, Basel, Switzerland RSS E-Mail Table of Contents Alert
Back to Top